# The Impact of International Student Mobility on Multicultural Competence and Career Development: The Case of Students from Latin America and the Caribbean in Barcelona

Robert G. Valls-Figuera [1,*], Mercedes Torrado-Fonseca [1] and Judith Borràs [2]

1 Department of Research and Diagnostics in Education, Universitat de Barcelona, 08007 Barcelona, Spain; mercedestorrado@ub.edu
2 English Department, Universidad de Diseño y Tecnología, 28016 Madrid, Spain; judith.borras@esne.es
* Correspondence: rgvalls@ub.edu

**Abstract:** Every year, an increasing number of students decide to study abroad in non-English-speaking countries, and Spain is recently a very popular destination within Latin American and Caribbean (LAC) communities. This paper attempts to deepen our knowledge of the impact of international student mobility (ISM) on students' multicultural identity and career development. To do so, the experiences of 10 LAC students who completed a one-year degree programme in Barcelona are examined. Semi-structured interviews were employed to evaluate the international experiences one year after their completion. During their post-mobility interviews, participants were able to reflect on the ISM through a structured dialogue that allowed them to analyse the experience from a distance. Findings indicate that the ISM helped them to grow personally and professionally and, one year after the stay, they are aware of this evolution. They show an increase in their self-confidence, and they see the experience as an opportunity for personal maturity. This suggests that universities should consider the importance of offering guidance to these students when they end their master's degree and are considering their plans for the future.

**Keywords:** internationalisation; international student mobility; career development; multiculturality; master's degree; higher education

## 1. Introduction

Considering the necessity for people in the XXI century to develop different competences that will allow them to solve problems and come up with sustainable solutions to worldwide issues, a focal point has been put on international student mobility (ISM) research because taking part in an international experience abroad could be key for the development of such competences [1]. Globalisation and internationalisation have become a defining issue for higher education institutions (HEI) in particular and society in general, and the number of investigations on the impact of ISM is only growing. Studying abroad can bring many benefits, not only to those students who embark on the experience, but also to both their origin and receiving countries. That is why it is necessary to learn more about these potential benefits so that international mobility policies can be written within the framework of internationalisation strategies created by different governments, HEI stakeholders, and policy makers.

A few investigations on the topic have focused on the impact that ISM can have on students' cultural, personal, and professional development [2–5]. In particular, researchers have tended to examine participants' professional development, their understanding of cultural differences, their language acquisition, their personal learning and global thinking, and their development of intercultural competences [4,6–12]. Nonetheless, there are still a few gaps in ISM research. In a recent review of the literature, Lee [13] asserted that students' perceptions about international mobility once the experience is completed remain

unexplored. Another gap in the literature is that studies normally investigate short-term programmes (one or two semesters, i.e., credit mobility), while longer programmes (one or two academic years, i.e., degree mobility) have received less attention. Finally, it is common to gather rather homogenous groups [14–17], which is why it is necessary to raise the voices of those groups of students that have been alienated from ISM research, as is the example of Latin American and Caribbean (LAC) students.

The present paper attempts to shed some light on some of the methodological limitations of ISM that have been highlighted by the previous research [4,5,18–20]. As previously stated, ISM research has often focused on Western populations, while other regions have remained unexplored [18,21] because of their intersectional [22] or unequal [23] nature. Moreover, long-term degree programmes have received particularly less attention within the ISM literature [24]. The present investigation examines a group of LAC students one year after the completion of their master's (MA) degree abroad. The study adopts a qualitative methodology in an attempt to understand the impact of international mobility on the students' perceptions of their cultural, personal, and professional development as well as the impact that the experience had on their multicultural competence growth and career development.

### 1.1. Cultural and Personal Impact of ISM

In the present study, the concept of "multicultural competence" is defined as the ability to understand and respect different cultural contexts and points of view, and includes openness to new ideas and ways of thinking [6]. Multicultural competence has been examined using quantitative [1,7,10,25–28] and qualitative approaches [8,29].

Research has centred its attention on the acquisition of key or transversal competences such as linguistic, intercultural, creative, and adaptability skills [8,30,31], which go beyond the more technical or professional skills that are acquired through (academic) training [32]. Personal growth has also been analysed regarding greater maturity, global commitment, and autonomy, among others [3,33].

In one of the largest projects on the topic, Farrurgia and Sanger [6] delved into the experiences of over 4500 alumni of US higher education institutions who had completed an international mobility between the academic years 1999–2000 and 2016–2017. They investigated the relationship between studying abroad and the development of skills related to employability and professional development by using a large list of soft and hard skills often required by employers. Most respondents declared having developed a wide range of cognitive, intrapersonal, and interpersonal aptitudes. In the same vein, Jon and Fry [3] and Petrie-Wyman et al. [34] established that international experiences contribute to a greater understanding and development of internal (e.g., self-awareness, risk assumption, open-mindedness, and attention to diversity) and external abilities (e.g., global awareness, intercultural competence, and collaboration among cultures).

Maddux et al. [35] put together a meta-analysis with the aim of confirming the impact of ISM on participants' intrapersonal, interpersonal, and organisational skills, considering creativity, psychological adjustment, intergroup bias, confidence, morality, leadership efficacy, and individual/professional performance. They proposed a model according to which the most profound multicultural experiences generate different integrative processes that transform participants' intrapersonal cognition, while wider multicultural experiences activate participants' comparative processes that influence their interpersonal attitudes and behaviours (i.e., cognitive processes that help people to integrate different cultural values and perspectives). It must be noted that Maddux et al. [35] highlighted that the aforementioned model works only when the overall perceptions of the experience are positive. In another study, Maharaja [8] investigated the personal diaries of 150 international students, and she concluded that the ISM contributed to a better understanding of the participants' own culture and that of others and to a greater self-confidence and global mindset. Similarly, the Study Abroad for Global Engagement (SAGE) project examined a group of 6391 US alumni who had completed an ISM between 1963 and 2005. A positive

impact was detected on five dimensions of global engagement (i.e., civic engagement, knowledge production, philanthropy, social entrepreneurship, and voluntary simplicity—the latter being the effort made to live a more modest and simple life), as well as on their educational and professional choices post-stay [33].

Nowadays, research also focuses on the development of students' global competences and the impact that this can generate not only on the country of origin but also on the networks and intercultural contacts that are created within the context of a more global education [36,37]. As an example, the international students in Rakovcová and Drbohlav [38] had integrated new values that they then spread among their own societies once they came back to their home countries. Returning home after immersion in another culture is often characterised as more difficult than the initial transition [39]. It is during the re-entry transition that individuals reflect not only on what they have experienced abroad, but on the new perspectives they have gained about their life at home [40,41], which makes them reassess their past values and beliefs [42]. In a study including participants from Latin America, Silva-Peralta and Rompato [43] indicated that ISM entails some relearning, which takes place because students have to overcome many obstacles during and after their sojourn. Hence, international students present an increased awareness regarding these challenges and an enlargement in their problem-solving skills, which leads to a more open mindset, greater knowledge of the self and their profession, more autonomy, an increased capacity to find solutions to different problems, and a more critical and integrated view of university training and society in general.

Considering the importance of the academic context, the theory of transformative learning, developed by Mezirow and employed by different researchers, allows us to approach the complexity of international students' transitions in an attempt to understand and explain how they may experience significant changes in their beliefs, values, and perspectives through their learning experiences [42]. This generates an impact on the cultural learning of international students [44–46]. In this sense, different aspects such as the curriculum, the coherence between values and rules, learning environments, and classroom ambiance have been examined. Paige et al. [33] highlighted that the nature of the programme can have a positive impact on students' development of their global competence. In other words, factors like the intensity of the educational experience abroad, the duration of the programme, or the students' active participation may directly impact participants' global competences.

Relevant findings are also found within GLOSSARI, a project about Georgia's university system (USA), which suggests an evaluation of international programmes abroad not only in relation to institutional indicators (e.g., enrolment and resources to support student mobility), but also concerning an examination of the acquired knowledge [31,32]. Initially, the project aimed at comparing different cohorts of students who went abroad or stayed at home. Different academic (programme and support) and mobility (length of stay and destination) variables were taken into account when examining how those learnings that happened during the ISM (in relation to functional knowledge, global interdependence, cultural relativism, verbal perspicacity, global geography, interpersonal adaptation, and cultural sensitivity) had impacted the participants' academic results and their subsequent professional trajectories. The results confirm the impact of ISM on all the examined areas.

### 1.2. Career Development in ISM

The impact of ISM on participants' professional development has been examined with a focus on the results, in terms of employment, and/or the process of employability and career management. Savickas [47] points at the importance of building a career based on the creation of an occupational identity. The author emphasises the relevance of reflecting upon one's professional trajectories in order to reach a significant career. Savickas et al. [48] explain that, in the current professional market, people need to be able to (re)adjust and adapt to new and changing circumstances, which requires constant learning and development.

Studies like that by Jon et al. [2] show that people who study abroad develop not only a greater understanding of themselves, but also the ability to explore, discover, and move forward when facing different challenges and working on their (future) careers. In this sense, the authors state that ISM helped participants to know more about themselves, to experiment within the context, and to face different problems, all of which gave form to their professional careers. The relationship between ISM and the labour market has been analysed from different perspectives: the transition to the professional market, career advancement, and employers' perspectives [5,49–54]. In a systematic review, Waibel et al. [55] found that, generally, participants believe that studying abroad improved their transition to the work market. In this line, Roy et al. [5] declared that participating in an international experience increased the chances of finding a job.

Different investigations have examined the relationship between mobility and subsequent employability as perceived by employers [52,56,57]. In general, findings suggest that having some international experience is appreciated by employers. This information is important for both home and host countries, especially when it comes to developing countries that are currently investing in human training [45,52,58]. Studies like those by Trooboff et al. [59] and Jacobs [56] confirm the value of ISM. Employers seem to take into account people's international experiences when making recruitment decisions. In particular, Jacobs [56] explains that the ISM gave an advantage in the labour market to participants in the study (Indian returnees). Singh [60] and Singh et al. [61] conclude that their Chinese participants who graduated from foreign institutions obtained some human, cultural, psychological, and identity capital that helped them find jobs once they returned to China.

Altogether, it is important to note that, apart from developing different intercultural competences, the challenges undergone during an international experience contribute to the acquisition of key competences that are often required in the labour market. Geyer et al. [30] compared a group who studied abroad to one that stayed at home, and they confirmed that being abroad had a significantly positive impact on the participants' leadership skills and professional aspirations. Ruth et al. [62] conducted a study with graduate students who declared that the mobility influenced different factors related to academic and professional success (e.g., professional connections, key competences, global view, and personal growth).

Researchers have also investigated the impact of mobility in terms of the process; that is, how the ISM influenced participants' professional and vital aspirations and expectations [2,5,19,63]. Paige et al. [33] examined the long-term impact of a short-term experience in relation to the participants' professional choices. The authors found that participation in ISM influenced the students' subsequent decisions on occupation and career. Nonetheless, even though a short-term (credit) mobility may influence students' decisions when looking for a new position, it has not been associated with greater employability or larger salaries in the long run, which contrasts with results found in relation to degree mobility [64,65]. That is why, as Sadeghi et al. [24] rationalise, it is very important to differentiate credit mobility (one or two semesters abroad) from degree mobility (at least one year abroad) when revising the findings related to ISM and the labour market.

Other relevant factors, such as opportunities for employability development in the host country as well as orientation and accompanying actions, have also caught researchers' attention. In an investigation with MA students, Fakunle [15] points at the responsibility of the receiving institutions to facilitate opportunities for employability development during ISM, as this could be key for students. Nilsson and Ripmeester [57] confirm that, from an employment perspective, it is sometimes difficult for students to transfer the knowledge acquired during the stay into their labour activities. Hence, this new learning is, many times, not used to its full potential. Pham [51] and Fakunle [66] state that there needs to be a stronger relationship between the university and different professional organisations, as well as a reassessment of university programmes, so that students are provided with enough resources and opportunities to plan their professional trajectories in the short and long run.

In sum, as pointed out by Silva-Peralta and Rompato [43] and Ruiz et al. [67], ISM entails a relearning that brings students to embrace new ways of understanding their reality and necessities. Additionally, it seems that ISM helps them to become globally prepared citizens who are free and critical and have a sense of social responsibility and just enough competences to manage the competitive labour market efficiently. Nonetheless, as previously stated, there are still a few gaps in the ISM literature [4,5,20], which are considered in the present paper. The objectives are to examine the impact that a one-year MA degree programme abroad had on a group of LAC students concerning (1) the development of their multicultural competences and (2) the construction of their professional careers. The following research questions guided the study:

RQ1: What are the competences and knowledge acquired during the ISM?

RQ2: How did the ISM transform students personally and in relation to their social values?

RQ3: How did the ISM influence the perspectives and development of their careers?

## 2. Materials and Methods

This paper presents the qualitative results of an investigation in which an ISM is analysed one year after it ended. A case study design has been adopted with the objective of understanding how the ending of the experience impacted international students' (present and future) life in a holistic manner.

### 2.1. Participants

Participants in this study come from a large pool of (national and international) students who completed an online questionnaire upon the completion of an MA at two different Catalan universities (for more information, see [68]). One year after the end of the MA, the international students were asked to voluntarily participate in an online interview with the objective of grasping a more robust understanding of their international experiences. Twelve international students gave their consent to participate in the interview and, after several communications, ten students were finally interviewed (n = 6 female, n = 4 male); they were selected with the objective of having a variety of countries of origin, ages, genders, types of mobility, and types of MA. This allowed for documentation of their differences, similarities, patterns, and peculiarities.

The 10 participants come from different Latin American and Caribbean countries, and they completed an MA degree that lasted for 1 academic course in Barcelona. The MAs had a research (n = 5), professionalization (n = 3), or mixed (n = 2) nature. For the selection of the MAs, only programmes having more than 5% international students registered were selected for this study. More information about the participants can be found in Table 1.

**Table 1.** Participants' information.

| Student | Country of Origin | University | Type of Master | Gender | Age | Type of Mobility |
|---------|-------------------|------------|----------------|--------|-----|------------------|
| S01 | Chile | UB | Research | Female | 42 | Individual |
| S02 | Puerto Rico (USA) | UAB | Professionalization | Female | 25 | Individual |
| S03 | Mexico | UAB | Professionalization | Male | 31 | Individual |
| S04 | Ecuador | UAB | Professionalization | Female | 27 | Individual |
| S05 | Paraguay | UB | Research | Female | 31 | Joint (with son) |
| S06 | Argentina | UAB | Mixed | Male | 29 | Joint (with partner) |
| S07 | Chile | UB | Mixed | Male | 34 | Joint (with partner) |
| S08 | Venezuela | UB | Research | Male | 24 | Individual |
| S09 | Dominican Republic | UB | Research | Female | 37 | Individual |
| S10 | Chile | UAB | Research | Female | 29 | Individual |

After their ISM, six participants returned to their home countries, while four remained in Spain for different personal and professional reasons. It must be noted that none of these

four participants had initially planned to stay, but they found the chance of doing so during their ISM and they took it.

### 2.2. Instruments and Data Collection Procedure

This study adopted a case study, qualitative methodology that allowed the researchers to grasp the voices of the protagonists through their own narratives, in which they describe the different factors that facilitated or complicated their transition after their stay [69]. A semi-structured interview was created to interview students individually. The different questions that guided the interview were created based on a theoretical analysis of the literature, and they were used to open a space in which participants could build on their lived experiences. Interviews were completed individually, and they took place online one year after the completion of their MA degrees. All interviews were carried out in Spanish, and they lasted for an average of 75 min.

Considering the objectives of this paper, the analysis and results focus on the final episodes of their ISM. The different dimensions and subdimensions as well as the categories and meanings that emerged can be found in Table 2.

**Table 2.** Description of the categories and subcategories analysed.

| Dimension | Subdimension | Category | Meaning |
|---|---|---|---|
| **Ending of the mobility project** | Post-mobility trajectories | Type of trajectory | Given the discontinuity and continuously changing nature of the students' experiences, different types of trajectories emerged among the participants attending to the combination of three elements: geographic location, professional situation, and project's consolidation. |
| | | Reverse (culture) shock | Analysis of those situations that generate stressful situations and are perceived as barriers among international students upon return to their home countries |
| | Evaluation of the impact | Professional development | Understanding the impact of the ISM in relation to students' employability and their development of technical and professional skills |
| | | Personal development | Understanding the impact of the ISM on students' personal changes |

### 2.3. Data Analysis Procedure

As previously explained, the semi-structured interviews took place one year after the completion of the students' ISM. Their trajectories and lived experiences were examined together with their perceptions in relation to the personal and professional impact that this experience had on them. The data from the individual semi-structured interviews were first transcribed verbatim by the first researcher. Then, the software NVivo v. 11 was used for the analysis of the data. This allowed for faster, yet accurate, management of the data, an increase in the coherence of the different analytical procedures, and a more transparent investigation process [70].

From the literature and content analysis, different inductive and deductive subcategories emerged (See Table 2). These were used to group the students' comments through a thematic analysis. For their description, three post-mobility defining elements were highlighted: (a) geographical location and changes throughout the ISM, (b) professional situation after ISM, and (c) the mobility project (finalised or in process). The intersection between these three elements shows the complexity of the trajectories and their evolution. Moreover, content related to their professional development was also collected: the progress

in the creation of their careers, the different transversal competences that they worked on during their ISM, and a reflection of their professional identities. From a personal perspective, we found different subcategories related to their intercultural development, their social commitment, their adaptability, and the autonomy that a stay abroad has given them.

## 3. Results

In the semi-structured interviews, students describe the learning and changes they have gone through as well as the impact of the ISM on their professional projects. The present section is divided into two main parts to answer the different research questions posed in the paper. In Section 3.1, findings are related to the development of the participants' multicultural competence and the impact of their (international) social context. A description of the reverse culture shock that some students perceived when returning to their home countries is also presented in Section 3.1. In the second section, the importance of the MA as a socialization context is analysed together with the changes in the teaching–learning models that promote the reflection of the participants' identities as potential education professionals. Finally, in Section 3.2, we examine the benefits of ISM in terms of the development competences that will be key for the participants' creation of their own professional careers. The participants' narratives have been interpreted in light of the biographies of each participant. All of them perceive the mobility experience as positive, and in particular, they highlight the impact that it has had on their professional careers. It must be noted that, considering that the language of the present paper is English, the excerpts in the results section were translated from Spanish to English.

### 3.1. Learnings throughout the ISM: Development of Participants' Multicultural Competence

All students have the same positive perceptions regarding their **multicultural competence development**. They feel that they are more capable of understanding cultural differences and that they are more accepting of them. S02 suggests that elements like "adaptability, wider knowledge, and multiculturality" are key. This is also explained by S04:

> *"The fact that I am being trained internationally [. . .] you are with lots of people [. . .] in Ecuador, you don't mix yourself with as many cultures as there are here in Barcelona; you open your mind to the world and realise that, in fact, we are all the same [. . .] for me, the benefit is that you open your mind and your culture."*

In MA degrees, as compared with undergraduate ones, students are usually expected to be more autonomous. Oftentimes, training is more specialised and implies a higher amount of autonomous work. In these cases, students are no longer simple classroom spectators, but they are given a more active role, which requires them to be ready to build on new knowledge, many times through collaborative methodologies. Many of our participants perceive this as a challenge, although they also accept that this challenge made them advance in their knowledge. In fact, the advantages of **collaborative work** are repeatedly highlighted during the interviews because they helped the students to develop their intercultural competences. This is reflected in this quote by S02, who comes from Puerto Rico and pursued a professionalization MA:

> *"You have to get used to all countries because we are used to working with our own group all the time, and we don't understand other cultures. At least here we are working with Latin American and other countries, and that's like being able to have a wider vision. Doing this MA, I could acquire this ability of being able to understand cultures of other countries."*

Students value the **relationships** they developed during their MA greatly and, in some cases, they maintained them after the completion of the programme. This created a very interesting network for them. Many refer to the MA as a space for intercultural

encounters that favoured adaptation and learning. According to this Argentinian student, whose trajectory includes law studies and some previous experience in Barcelona:

> *"My year was very multicultural, there were many countries. And, even if there were many differences—Argentina is very different from Chile, very different from Colombia, different from Ecuador, it's very different—it was very advantageous, we learnt a lot from ourselves."* (S06)

In the same vein, the group created a **multicultural atmosphere** that generated learning at many different levels. For some of them, this learning was something practical because they learnt about new realities and ways of working. This was particularly the case within professionalization MAs.

The impact of multicultural experiences clearly contributes to the development of **students' social skills**. S09 says: *"Finding myself among these people coming from such different cultures made me a bit more open, because I'm quite shy."* Moreover, it favours the creation of an international network that helps them to move around new contexts, especially among those people who go for careers with an international look.

In other cases, students' narratives highlight the impact of the experience in relation to their more **global commitment with their country**, as is the case of student S07, from Chile: *"I don't know if I'd like to stay here or go somewhere else, or to Barcelona, but I feel that here [in Chile] there is a lot of work to do."* This is, in fact, a path that some have already started taking:

> *"We can do things and improve as a society. I'm teaching at a university degree level to make 20-year-olds feel motivated. For them to listen to someone who went there and did a good job just so that they could see that they can do similar things. I feel blessed and I want to share this blessing in order to tell them that they can do it, that we can improve by living other experiences."* (S03)

Participants delve into aspects related to **social values**, and **differences among cultures** become more evident. Among such differences, the following are particularly underlined: social politics, perception of security, relationships between different social classes, women's role, relevance of public spaces, work hierarchy, and the importance of aspects like punctuality. The interviewees admire the quality of life in their host country and the quality of (public) services inside and outside university, and this clearly contrasts with their previous experiences. They seem remarkably surprised by the use of public spaces such as parks or museums and the fact that most services are free. Through these new lenses, some participants re-examine their own countries.

The international experience lived in Barcelona, a cosmopolitan city that is open to the world, elicits some reflection on the participants' own societies. The differences between the home and host country become evident, which generates mixed feelings once the ISM is over and students have to go back to their countries of origin. The reintegration into their own societies is not easy because students perceive changes in their identities and in their understanding of life and society. They highlight the impact derived from having learnt about so many other cultures and societies and the complexity of the **readjustment** process. In their post-mobility narratives, students talk about having a negative perception of their home societies, which are seen as resistant to change. This is exemplified by a Mexican student whose return had provoked feelings of frustration towards his own society:

> *"I could feel the differences when I got back. Being in a society in which differences are not so evident, when I came back, I could perceive such differences even more. It makes you want to stay there [in Barcelona] [. . .]. You come back and you see the full scan of what your society is, then I understood. Not only I felt nurtured by my trip, but also I felt a bit sad when realising what my own country really is. I had never experienced what I am living abroad, and it's an intense experience."* (S03)

It must be noted that, even though all of them have a positive perception of the experience, there are a few differences in the emphasis each participant gives to their own

learning or to the different competences they have acquired. Throughout this subsection, we have seen quotes of different intensities regarding the students' degree of transformation. Nonetheless, all of them argue that the personal changes and the changes in the way they view things can be attributed to the international experience in Barcelona. Altogether, it seems that students' experiences tend to be valued as more positive when students' initial expectations were satisfied.

*3.2. Change and Reflection Concerning Their Professional Projects: Impact on Their Career Development*

The ISM is perceived as extremely valuable by all participants. These international experiences entailed a reflection about their careers. In other words, the ISM greatly influenced their decisions and (future) professional trajectories. Considering the participants' employment situations at the moment of the interviews, the following significant changes can be observed: of position (S02, S09), of organisation (S03, S04, S05, S10), of organisation and sector (S01, S07), and of professional career (S06, S08). As an illustration of such changes, a Chilean student (S07) put an end to his teaching career to start working in a sector that allowed him to have more interaction and impact on society. With this objective, he created a nongovernmental organisation (NGO) to tackle environmental education. Another example is that of the Ecuadorian participant (S04), who decided to extend her stay in Barcelona because she had the chance of working at the same company where she completed her MA practicum and starting an international career:

> "I have considered going back to Ecuador in the long run, but just after I get some professional experience abroad. . . maybe I'd go back in three, four, or five years, and I'd have my own communications agency that could have the credentials of a multinational company."

In their narratives, participants indicate that the MA was their main **context for socialization and learning**. They encountered a new way of learning and a different way of understanding university and professional training. The **new teaching model** caught their attention from the beginning, and the new methodologies raised some tension and ideological confrontation in relation to some key factors such as the role of the teachers and their prestige, flexibility, autonomy given to students, or assessment processes. Nonetheless, this re-evaluation of their own (previous) models makes them value the many benefits of the new one, as S01 points out:

> "Despite realities in Latin American countries normally seeming slightly similar, I went through a huge internal change because at Universitat de Barcelona, here, I learnt about how public education was in my country, and in Latin America in general."

Generated through the reflection of a comparison between new and old models, this vision is shared by those students who had some previous professional experience in the education sector. The comparison between their host and home countries made them value this change as positive, highlighting the importance of the teachers' identity:

> "There [in Spain], they are teachers and researchers, and you can tell that they dedicate themselves to that and they enjoy it. Here [in Argentina], many teachers just do it to have it in their CV, but they don't have a vocation for teaching, and that's something you can tell." (S06)

In relation to this **teaching identity**, participants were surprised by the relationships and interactions they created with their teachers abroad. This also prompted some reflection about their own professional performance, especially among those students within education fields, who highlight the closeness, the equitable relationship, and the individual feedback they received from lecturers, and also the fact that their own feedback was actually taken into account by teachers. All of this was particularly remarkable for participants in the present study because they considered their teachers professionals in the field. They had read papers and books written by their current teachers and they admired them, so they expected a more hierarchical relationship. Hence, their ISM transformed their perceptions

of teachers as having all the authority in the classroom into teachers simply being one more actor in the learning process. Students point out the respectful and integrative stance taken by teachers and the equitable relationship among teachers and students, all of which made them feel accompanied:

> "*I came out of class and requested a meeting. Then, you talked to the teacher and solved the issue. I guess that this availability that teachers had is a characteristic of full time MAs, the availability they had for us surprised me because in Paraguay teachers work under precarious conditions.*" (S05)

Some participants show a more utilitarian vision, analysing the impact that the ISM clearly had on their **employability benefits**. The MA provided them with different technical and professional skills as well as with conceptual and methodological tools that helped them with the **understanding and development of their professional careers**. S02 argues that "*what the MA did was opening my mind and expand my knowledge [. . .] having a wider vision of what my career is.*" Moreover, the newly acquired knowledge was easily transferred to their professional life. This is the case of S06, an Argentinian student who learnt about research and research methodologies, which were key for his job. After the MA, he was able to conduct his own investigations.

The three students who went for a professionalization MA highlight the development of key competences like autonomy, assertiveness, self-determination, and the assumption and understanding of risk when making decisions and starting new projects. A few other students see themselves as more proactive, confident, and effective in their work environment owing to different competences they acquired during their stay. Their words allow us to glimpse this empowerment, which underlines the need to discuss different issues:

> "*Now I coordinate the social sciences area and something that had never crossed my mind was to sit and have a debate with other teachers [. . .] This seems really interesting because it opens new possibilities.*" (S05)

As explained by this Mexican student, two other key competences acquired by the students during their programme abroad were planning capacity and management of their own professional projects: "*that's what the MA taught me: to define my own targets and objectives in the short, medium, and long term, and to put them into practice*" (S03). S10 also mentions that the ISM in general and the MA in particular helped her develop her capacity to manage and control her emotions.

In one way or another, all participants note that their **professional growth** and the development of different transversal competences was beneficial for them at both a personal and professional level. Implicitly, the experience opened a space for reflection about their careers and their own identities. In other cases, this is expressed at a more profound level, as is the case of this Chilean student regarding her professional trajectory as a teacher: "*This educational change was not external, it was internal in my case [. . .] at Universitat of Barcelona I've got to know more about me and about the public education in my home country*" (S01). This personal transformation goes beyond the knowledge they acquired during their ISM, as stated by S07, who says that he could have completed the MA in his country (Chile) and obtain a new medal for his CV, but that would not have changed his view of things or put on some new lenses, which was the leap towards greater change.

The ISM served as a catalyst for the students, and it brought them closer to new professional aspirations, changed their priorities and objectives, and made them revise and redirect their professional projects. In any case, the MA has brought a new meaning and a new professional projection, and it has given them a more global vision and a stronger commitment to their own countries. The reflection about their own personal and professional growth has made them aware of their growth, with quotes that refer to self-confidence and the ISM as an opportunity for personal maturity.

## 4. Discussion and Conclusions

The present paper aimed at examining the impact of completing a degree programme abroad one year after the completion of the experience. It examined participants one year after the completion of their MA programme abroad, which brought the opportunity to analyse their lived experiences with a temporal perspective. This allowed time for reflection and evaluation of the experience. Moreover, participants are all LAC international students, which provides some insight on this rather under-researched group. Results show the knowledge and competences acquired by the students in terms of multicultural development and reflection towards their professions and careers.

In line with previous studies, participants agreed on the benefits that the international experience brought [8,30,32]. Not only did the ISM help them acquire different technical and professional competences, but also a variety of key and transversal skills (the capacity to understand and operate in different cultures, creativity, and adaptability) and soft skills that are fundamental in our globalised world [33,56].

Degree mobility at an MA level seems to have improved the students' self-confidence, problem-solving skills, and expectations related to self-efficacy. Moreover, it has helped them when creating new contacts and international networks. Hence, it can be asserted that completing an MA programme in a foreign country is likely to have a positive impact on students' employability [50,61,65]. Participants in the present study declare that studying abroad provided them with more opportunities in the labour market and a better professional situation. Although it must be noted that the destination country and receiving institution may play a role in this, it seems that completing an international experience gives participants the chance of finding a (better) job in their own countries. This finding confirms what previous studies have found regarding the competitive advantages that ISM provides [5,12,35,56,59,60].

Research shows that employers value not only people's technical knowledge and skills, but also the transversal competences that can be acquired within an international context and that oftentimes go beyond academic development [30,43,62]. Results in our study indicate that being abroad can bring significant changes in terms of adaptability and resilience, which is of help when it comes to managing tasks and the (expected and unexpected) challenges related to the professional career [3,8]. Participants' employability skills strengthen when they are placed in multicultural contexts in which they must learn to cope with difficult situations and new and challenging environments without the support of their support network.

The students' narratives also present significant personal changes in relation to their maturity, reflection, and social commitment [3,29,41,68]. In some of these narratives, the experience is portrayed as having contributed to opening students' minds and to creating a stronger social commitment towards their respective countries [38,43]. This finding is in line with Nada et al. [42] and the theory of transformative learning. The personal changes described by students are a consequence of having been abroad, which has made them question and revise their ideals and assumptions. In many cases, the reverse culture shock that is experienced when returning to their home countries [40] causes a feeling of frustration towards the participants' own societies [41,71]. Being aware of the changes in their identity, values, and understandings makes some international students opt for new ways of thinking and for new approaches to how to see the world and how to be more committed to their countries.

One last finding is related to how the experience influenced participants' reflections on their professions and the construction of their careers. The present paper provides evidence that ISM can lead participants to reaching a greater understanding of their own professional selves [38,41,51]. Being exposed to new educational environments, opportunities, models, and sociocultural rules created new interests in participants [38,72], which made them see their professional paths as trajectories that are fluid and in continuous change. Participants expressed how their professional plans are now open and under construction, and some

of them have started gaining interest in more international careers [3,5,19], or, at the very least, showing a more global thinking.

In a nutshell, the protagonists' narratives highlight how the international experience impacted them in a very positive and global way. The ISM was a stimulating experience that gave them new professional aspirations, made them change their priorities and objectives, and required a revision and reorientation of their professional trajectories and projects. The MA gave the participants meaning and professional projection. Additionally, some have increased their global commitment to themselves and their countries. Considering the potential advantages that ISM can provide to both host and home countries, and the potential benefits that it can bring in terms of participants' personal and professional development, investigating the impact of ISM should be a core priority in the internationalisation agenda for researchers, practitioners, and policymakers.

Likewise, it is necessary to promote different mobility policies within the internationalisation strategies of countries and institutions around the globe. It is important for higher education teachers and staff to understand and embrace the necessity of developing participants' professional and personal identities so that, together, they can guide students through their degrees. Finally, peer contact is also of utmost importance, especially when such contact is with people of different nationalities and cultures, because it favours the development of students' multicultural competence [27]. Altogether, the role of the higher education institutions is fundamental when it comes to designing actions that facilitate cultural experiences and employment opportunities [66] among local and international students. This should be accomplished using a perspective based on equity and social justice [73] that allows equal access to mobility opportunities among all students.

Despite the paper providing new insights into the literature, it is not without limitations. First, only 10 participants are part of the analysis, and they do not represent all LAC countries. In future research, it would be interesting to have participants from other countries, to examine whether results would be the same. Although an innovative aspect of this study is that it conducted follow-up interviews one year after the stay, researchers should consider carrying out a more longitudinal investigation that also takes into account participants' views right after the completion of their degrees. In other words, future studies would do well in monitoring the students post-mobility.

**Author Contributions:** Conceptualization, R.G.V.-F., M.T.-F. and J.B.; methodology, R.G.V.-F. and M.T.-F.; software, R.G.V.-F. and M.T.-F.; validation, R.G.V.-F. and M.T.-F.; formal analysis, R.G.V.-F., M.T.-F. and J.B.; investigation, R.G.V.-F. and M.T.-F.; resources, R.G.V.-F. and M.T.-F.; data curation, R.G.V.-F. and M.T.-F.; writing— original draft preparation, R.G.V.-F. and J.B.; writing—review and editing, R.G.V.-F., M.T.-F. and J.B.; project administration, M.T.-F.; funding acquisition, M.T.-F. All authors have read and agreed to the published version of the manuscript.

**Funding:** This research was funded by the Economy and Competitiveness Ministry, Spain. EDU2016-80578-R: funded by MCIN/AEI /10.13039/501100011033 and the European Regional Development Fund: a way to make Europe.

**Institutional Review Board Statement:** This research adhered to the standards of the Social Sciences of the Ethical Committee of Experimentation of the University of Barcelona (Spain).

**Informed Consent Statement:** Informed consent was obtained from all subjects involved in the study.

**Data Availability Statement:** The data presented in this study are available on request from the corresponding author. The data are not publicly available due to privacy and ethical restrictions.

**Conflicts of Interest:** The authors declare no conflict of interest.

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
