# Peer review of "The Impact of International Student Mobility on Multicultural Competence and Career Development: The Case of Students from Latin America and the Caribbean in Barcelona"

_education, doi:10.3390/educsci13090869_

Round 1
Reviewer 1 Report
The conducted follow-up interviews are an excellent effort and the authors are encourage to report future studies.
Reference 1, has a minor error, journal pages are 1-11.
Reference 4, the name of the first author should is Jon, J-E.
Please review all references, there might be more errors.
Author Response
Dear reviewer,
We appreciate your comments, revisions, and suggestions to improve our contribution. In order to clarify them we share our considerations.
Point 1: Please review all references, there might be more errors.
Response 1: All the references and their doi have been checked.
We appreciate your time and valuable input.
Reviewer 2 Report
I guess in-text citations in the abstract could be awkward. It would be better to revise the abstract without citations.
In the 2.1. Participants section, can you explain why the author(s) chose only 10 out of 12 students who agreed to participate?
The 3. Results section is very hard to read and follow the flow of the manuscript. It would be desirable to re-organize suing sub-headings or tables so that readers can follow the flow efficiently; otherwise, the discussion and conclusion may not be strongly supported by the organized findings.
Author Response
Dear reviewer,
We appreciate your comments, revisions, and suggestions to improve our contribution. In order to clarify them we share our considerations.
Point 1: In-text citations in the abstract could be awkward.
Response 1: In the case of including this first quotation, it was related to the fact that it is a methodological element that allows a better and greater understanding of the phenomenon. However, the sentence has been rewritten so that the reference does not have to be included.
Point 2: In the 2.1. Participants section, can you explain why the author(s) chose only 10 out of 12 students who agreed to participate?
Response 2: There has been a small error in the participants section and therefore the sentence has been rewritten. The interviewees were selected from those students who gave their consent to participate in a previous questionnaire; thus, of the 12 international students who gave their consent to participate in the interview, and after several communications, 10 students were finally interviewed. L230-231: Out of the 12 students who agreed to participate in these interviews, a total of 10 à Twelve international students gave their consent to participate in the interview and, after several communications, ten students were finally interviewed.
Point 3: The results section is very hard to read and follow the flow of the manuscript. It would be desirable to re-organize using sub-headings or tables so that readers can follow the flow efficiently; otherwise, the discussion and conclusion may not be strongly supported by the organized findings.
Response 3: In relation to the introduction of subtitles to facilitate the reading of the results, the same structure has been followed as in the theoretical part: for this, there is a first section related to the learnings throughout the international mobility and the development of multicultural competence (this first section in the results is entitled: "Learnings throughout the ISM: Development of participants' multicultural competence") and, secondly, the impact of international mobility on the development of the professional career (section entitled "Change and reflection concerning their professional projects: Impact on their career development").
We appreciate your time and valuable input
Reviewer 3 Report
its an important research with contribution
suggest minor correction
Author Response
Dear reviewer,
We appreciate your comments, revisions, and suggestions to improve our contribution. In order to clarify them we share our considerations.
Taking into account the idiomatic corrections, the text has been revised once again.
We appreciate your time and valuable input.
Round 2
Reviewer 2 Report
I still think that the result sections (3. Results) should be more divided using subheading below 3.1 & 3.2. I guess the 3.1 & 3.2 headings can be divided into smaller themes. Other than that all the comments were full integrated into the revision. Thank you.
Author Response
Dear reviewer,
We appreciate your comments, revisions, and suggestions to improve our contribution. In order to clarify them we share our considerations.
Following the comments made in the reviews, it has been decided to introduce a more detailed initial paragraph, explaining the categories of analysis present in the results section. In addition, in order to complete these, and taking into account the ease of reading the qualitative results, but without "breaking" too much with the structure established in the rest of the article, we have chosen to highlight (in bold) the topics analysed.
Proposal "In the semi-structured interviews, students describe the learning and changes they have gone through as well as the impact of the ISM on their professional projects. The present section is divided into two main parts so as to answer the different research questions posed in the paper. In section 3.1., findings are related to the development of participants’ multicultural competence and the impact of their (international) social context. A description of the reverse culture shock that some students perceive when returning to their home countries is also presented in section 3.1. In the second section, the importance of the MA as a socialization context is analysed together with the changes in the teaching-learning models that promote the reflection of the participants’ identity as potential education professionals. Finally, in section 3.2. we examine the benefits of ISM in terms of the development competences that will be key for the participants’ creation of their own professional careers. The participants’ narratives have been interpreted in light of the biographies of each participant. All of them perceive the mobility experience as positive and, in particular, they highlight the impact that it has had on their professional careers."
We appreciate your time and valuable input.